# A Review of the Recent Development in the Synthesis and Biological Evaluations of Pyrazole Derivatives

**DOI:** 10.3390/biomedicines10051124

**Published:** 2022-05-12

**Authors:** Oluwakemi Ebenezer, Michael Shapi, Jack A. Tuszynski

**Affiliations:** 1Department of Chemistry, Faculty of Natural Science, Mangosuthu University of Technology, Durban 4026, South Africa; ebenezer.oluwakemi@mut.ac.za (O.E.); mshapi@mut.ac.za (M.S.); 2Department of Physics, University of Alberta, Edmonton, AB T6G 2E1, Canada; 3Department of Oncology, Cross Cancer Institute, University of Alberta, Edmonton, AB T6G 1Z2, Canada; 4Department of Mechanical and Aerospace Engineering, (DIMEAS), Politecnico di Torino, 10129 Turin, Italy

**Keywords:** heterocycle, pyrazole, derivatives, synthesis, biological activities, recent development

## Abstract

Pyrazoles are five-membered heterocyclic compounds that contain nitrogen. They are an important class of compounds for drug development; thus, they have attracted much attention. In the meantime, pyrazole derivatives have been synthesized as target structures and have demonstrated numerous biological activities such as antituberculosis, antimicrobial, antifungal, and anti-inflammatory. This review summarizes the results of published research on pyrazole derivatives synthesis and biological activities. The published research works on pyrazole derivatives synthesis and biological activities between January 2018 and December 2021 were retrieved from the Scopus database and reviewed accordingly.

## 1. Introduction

Heterocycles are a fundamental and unique class of compounds; they account for over half of all known organic compounds and have a broad range of physical, chemical, and biological properties, covering a broad spectrum of reactivity and stability [1]. Furthermore, their synthetic usefulness as synthetic intermediates, the protective groups, the chiral auxiliaries, the organocatalysts, and the metallic ligands in the asymmetric catalysts in pharmaceutical agents have rendered them multiple units of interest. Among heterocyclic compounds, five-membered rings containing nitrogen atoms constitute a vast and differentiated group with a broad spectrum of biological activity [2,3,4]. The members of this group, such as pyrazole, imidazole, oxazole, triazole, tetrazole, oxadiazole, thiazole, and isoxazole, are particularly important antibacterial and antifungal agents [3,4,5]. The pyrazole ring is a five-membered heterocycle containing two adjacent nitrogen atoms. It is a moiety found in many molecules that possess many applications. Additionally, naturally occurring pyrazoles and their synthetic derivatives are well-known to have a broad spectrum of biological properties (Figure 1). In recent years, some of the FDA-approved and commercialized drugs, including patented ones, have been developed from pyrazole derivatives (Figure 2), which implies ample usage of these groups in new-fangled bioactive molecules. This review focuses on a concise overview of the pyrazole pharmacophore synthesis and biological activities reported between 2018 and 2021. Thus, it will serve as a helpful reference guide for researchers interested in the field. This review is loosely categorized into chemical synthesis and biological applications. The first section includes the synthesis of pyrazole derivatives, and the second section describes the biological applications.

## 2. Synthesis of Pyrazole Derivatives

### 2.1. Condensation of Hydrazine’s or Similar Nuclei with Carbonyl Functional Group Compounds

#### 2.1.1. Pyrazoles from Vinyl Ketones

The tandem reactions between amine-functionalized enaminones **1** and aryl sulfonyl hydrazine or tosylhydrazone derivatives **2** and **3** in the absence of a metal catalyst have been reported [6]. The synthesis of the substituted pyrazoles **4** and **5** occurred in water, TBHP, and NaHCO_3_, respectively. In addition, when alkyl-based sulfonyl hydrazine such as methyl sulfonyl hydrazine was incorporated, the reaction was not successful. Additionally, when the reaction was carried out in EtOH and DMF, all the analogs were obtained with a good yield. The expected products with a lower yield were formed by introducing double-ethyl functionalized enaminone (R_1_ = R_2_ = ethyl). (see Figure 1).

Wan et al. [7] reported a different synthetic route to forming-substituted pyrazole derivatives, including celecoxib (**7a**), mavacoxib (**7b**), and deracoxib (**7c**), respectively. The compounds were synthesized using enaminones and aryl hydrazines in ethanol with acetic acid. The reaction produced regioselective compounds with a high yield. Notably, compared to the other synthetic methods, using fluoroalkylated pyrazoles [8], β-diketones [9], and ynones [10], this method gives an excellent yield, a regioselective product. The synthetic route can be explored to synthesize pyrazole derivatives that are not easy to get from fluoroalkyl β- diketones. (see Figure 2).

#### 2.1.2. Pyrazoles from 1,3-Diketones

A silver-catalyzed synthesis of 5-aryl-3-trifluoromethyl pyrazoles using *N*’-benzylidene tolylsulfonohydrazides **8** with ethyl 4,4,4-trifluoro-3-oxobutanoate **9** as precursors has been reported [11]. The reaction involved consecutive nucleophilic addition, intramolecular cyclization, elimination, and finally, [1,5]-*H* shift. This led to trifluoromethylated pyrazole derivatives **10** with moderate to excellent yields (see Figure 3). In optimizing the product, the yield improved by increasing the reaction temperature to 60 °C, but increasing the reaction temperature above 60 °C resulted in a lower yield. The Cu(OTf)_2_ transition catalyst afforded 60% yield, while Fe(OTf)_3_ was unproductive. THF or dioxane gave a poor yield of the product compared to toluene. Meanwhile, K_2_CO_3_ was more effective than NaH, KO*t*-Bu, and NaO*t*-Bu. Additionally, the use of Me_2_phen as a ligand yielded the best performance (>99%), compared to using bpy or phen as a ligand (57% or 92%).

Poletto et al. [12] reported the one-pot synthetic strategy for synthesizing highly regioselective α-ketoamide *N*-arylpyrazoles **28**, the secondary *β*-enamine diketone, and arylhydrazines as precursors. Notably, the intermediate 4-acyl 3,5-dihydroxypyrrolone, produced in situ, went through nucleophilic substitution at C-5 by arylhydrazine. Afterward, heterocyclization occurred at the carbonyl carbon of the acyl group (see Figure 4).

#### 2.1.3. Pyrazoles from Acetylenic Ketones

The merging of substrates with five electron-rich heteroaromatic nuclei interacts with arylhydrazines with the carbonyl group and triple carbon bond. The cyclo-condensation of cross-conjugated enynones **13** with hydrazines has produced pyrazole derivatives **14** and **15** in good yield [13] (see Figure 5).

### 2.2. Dipolar Cycloadditions

#### 2.2.1. Pyrazoles from Diazoester

Fang et al. [14] reported the designed and synthesized polysubstituted 4-trifluoromethylpyrazoles using ketones **16** and trifluoroacetyl diazoester **17**. The ketones reacted with the terminal nitrogen atom of the trifluoroacetyl diazoester, followed by cyclization, forming 4-trifluoromethylpyrazole derivatives **18**–**27**. Meanwhile, the replacement of DBU with NEt_3_, NaO*t*-Bu, CsF, Cs_2_CO_3_, and Na_2_CO_3_ resulted in a poor yield (see Figure 6). Notably, dialkyl ketones did not afford the corresponding 4-(trifluoromethyl)pyrazoles **26** and **27**, using DBU and NEt_3_, which could be due to the relative lower reactivity of the α-hydrogen of dialkyl ketones.

Chen et al. [15] reported a synthetic approach for synthesizing polysubstituted 4-difluoromethyl **28** and perfluoroalkyl **31** pyrazole derivatives. The authors utilized a Lewis acid and base co-mediated reaction of perfluoroacetyl diazoester with ketones (see Figure 7). Catalysts such as Cu(OTf)_2_, CuCN, Sc(OTf)_3_, NiCl_2_, FeC_l3_, Fe(OTf)_3_, CoCl_2_, and ZnI_2_ were explored for the optimization of the reaction. Particularly, the Sc(OTf)_3_ catalyst displayed the best performance with 97% yield in the presence of DBU as the base. Meanwhile, the base, such as Et_3_N, _t_-BuOK, K_2_CO_3_, and K_3_PO_4_, did not give the expected product.

#### 2.2.2. Pyrazoles from Vinyldiazo Ketones

The synthesis of pyrazole-based triarylmethanes using 2-(1-alkynyl)-2-alken-1-ones and vinyldiazo ketones has been reported [16], as shown in Figure 8. The first stage of the reaction involved heating vinyldiazo ketones **32** in dichloroethane, which produced 1*H*-pyrazoles **33**, followed by a reaction with enynones **34** to produce pyrazole-based triarylmethanes **35**.

#### 2.2.3. Pyrazoles from Hydrazones

Zhu and colleagues [17] investigated the oxidative coupling reaction of phenylhydrazone **36** and maleimide **37** to synthesize pyrazoles derivatives. The reaction was carried out with CuCl as the catalyst, and dimethylformamide (DMF) was used as a solvent (see Figure 9 and Figure 10). In the reaction, it produced 12% yield in the presence of 20 mol% Cu(OAc)_2_ in dimethylsulfoxide (DMSO) at 80 °C for 2 h. Additionally, other Cu(II) salts did not enhance the reaction. The reaction yielded 86% when CuCl as the catalyst in DMSO was utilized, while Cu(I) salts, mainly CuOAc, CuBr, CuI, and CuSCN, led to product reduction. Moreover, catalysts such as Mn(OAc)_3_, Ag_2_ CO_3_, FeCl_3,_ and Pd(OAc)_2_ were inefficient.

An effective protocol for synthesizing pyrazoles derivatives **46** and **49** via an iodine-catalyzed reaction of aldehyde hydrazones with electron-deficient olefins has been reported [18] (see Figure 11). The transformation of the reaction produced a 35% yield in the presence of 20 mol% I_2_ and 3.0 equiv of TBHP in DMF at 80 °C. The solvents, such as CH_3_CN, displayed a moderate yield. When different oxidants were utilized, BPO displayed a superior performance up to an 81% yield compared to TBHP, K_2_S_2_SO_8_, DTBP, BTI, H_2_O_2_, and m-CPBA. The product was reduced by replacing molecular iodine with other iodides, such as NaI, NIS, or TBAI.

A facile one-pot, copper-catalyzed aerobic cyclization has been consecutively used to synthesize the pyrazole derivatives (**51** and **54**) by Fan and coworkers [19]. In this reaction, β and γ-unsaturated hydrazones were readily available substrates. While O_2_ acted as the terminal oxidant and economic Cu(I) salt was used as the catalytic agent, CuOTf showed the best performance compared to other employed catalysts such as CuOAc, CuBr, Cu(acac)_2_, CuOTf, and Cu(OTf)_2_ (see Figure 12 and Figure 13).

The intermolecular, thermally activated, and DBU-aided [3 + 2] cycloaddition of pyridin-2-yl-[1,2,4]-triazine dipolarophiles **55** with structurally varied 4-methylbenzenesulfonohydrazides **24** produced **57** as the major isomer with an excellent yield [20] (see Figure 14).

Zheng et al. [21] used a metal-free protocol to synthesize pyrazolylthienopyrimidines and other *N*-heteroaryl pyrazole derivatives **57** from α,β-unsaturated *N*-tosylhydrazones **103** and *N*-heteroaryl chlorides **58** under mild reaction conditions. The bi(heteroaryl) pyrazole derivatives were obtained in good to excellent yields (see Figure 15).

#### 2.2.4. Pyrazoles from Diazo Intermediates and Alkynes

Dimirjian and colleagues [22] developed a synthetic approach for synthesizing fused pyrazoles via an intramolecular reaction. Notably, 1,3-dipolar cycloadditions of diazo intermediates with alkynes **60** produced the spirocyclic product of pyrazole derivatives **62** (see Figure 16).

#### 2.2.5. Pyrazoles from Vinyl Sulfone

Dihydro-pyrrolo-pyrazoles have been synthesized through a cascade reaction involving cinnamyl azides and vinyl sulfones with moderate to good yields [23]. The protecting group, ethylene sulfone, can be removed by heating the product in pyrrolidine (see Figure 17). The reaction tolerated a range of solvents such as benzene, acetonitrile, methanol, 1,3-dichloroethene, isopropanol, and dioxane. However, dioxane with triethylamine as a base produced dihydro-pyrrolo-pyrazole excellently compared to dioxane and diisopropyethylamine (DIPEA) or diisopropanolamine (DIPA).

#### 2.2.6. Pyrazoles from Nitro-Olefins

The Rauhut−Currier cyclization reaction was reported for the synthesized trisubstituted pyrazole derivatives [24]. The trisubstituted tetrahydropyrano [2,3-c]pyrazoles were obtained from the domino Rauhut−Currier cyclization reaction. The reaction occurred between alkylidene pyrazolones and nitro-olefins (see Figure 18).

#### 2.2.7. Pyrazoles from Alkynes

A visible light-promoted cascade of Glaser coupling/annulation alkynes and hydrazines has been utilized to synthesize polysubstituted pyrazoles **73** and **75** [25] (see Figure 19). The replacement of CuI with CuCl, CuCl_2_, Cu(OTf)_2_, and Cu(OAc)_2_ in the reaction using maintaining Ru(bpy)_3_Cl_2_ as the photocatalyst did not improve the reaction product.

#### 2.2.8. Pyrazoles from Morita−Baylis−Hillman (MBH) Carbonates

Under mild reaction conditions, phosphine-catalyzed domino Morita−Baylis−Hillman (MBH) carbonates with diazenes yielded tetrahydropyrazole-fused heterocycles **78** and **81** with moderate to excellent yields [26] (see Figure 20). The substitution of *tert*-butyl (**81g**) with MBH carbonate was not reactive, probably due to its steric barrier. It is noteworthy that MBH carbonates derived from other aldehydes, namely benzaldehyde, did not function in this reaction.

##### Multicomponent Strategies

A one-pot multicomponent reaction for synthesizing bispyranopyrazole **86** derivatives using MMT K10 as a support heterogeneous catalytic system has been reported [27] (see Figure 21). Notably, the economic and environmentally friendly catalyst was recycled and reused five times in the reaction, and no lack of activity was observed.

Alizadeh-Kouzehrash et al. [28] reported new *N*-fused pyrazole derivatives **91** via an efficient one-pot multicomponent reaction. Using a cheap catalyst, namely 4-toluenesulfonic and ethanol, as a green organic solvent (see Figure 22), the best yields were reached in ethanol as a solvent under the reflux temperature, and in the absence of a catalyst, the yields percentage of reactions were reduced.

## 3. Miscellaneous

The synthetic route to JNJ-18038683 **83** and other fused pyrazole derivatives were reported by Dvorak et al. [29]. Two synthetic routes were employed to construct the fused pyrazole-azepine heterocyclic core (see Figure 23, Figure 24 and Figure 25). In the reaction path to pyrazole triflate **107**, a bi-phasic solvent system of toluene/water was optimal, and no triflate hydrolysis was identified. Meanwhile, the displacement of the BOC-protecting group was achieved by treating **107** with trifluoroacetic acid to give the free base of the amine. Subsequently, the free base was converted into citrate salt, and the salt formation was then performed with a free base and citric acid in methanol. The final recrystallization yielded clinical candidate **108** as a nonhygroscopic free-flow powder.

1*H*-pyrazole-5-amines were obtained from the microwave reaction of arylhydrazines **108** with 3-aminocrotonitrile **109** in moderate to excellent yields after 10 min of irradiation [30]. In addition, the reaction of phenylhydrazine **110** with α-cyanoketones **110** under similar conditions produced many functionalized 1*H*-pyrazole-5-amines. Meanwhile, the *m*-nitrophenyl and *p*-nitrophenyl-3-oxopropanenitrile substituents did not react with phenylhydrazine, even at a longer heating time. However, the reaction conditions tolerated other functionalized aromatic groups such as trifluoromethyl and methyl sulfone (see Figure 26).

## 4. Biological Activity of Pyrazole Derivatives

### 4.1. Anti-Inflammatory

Bhale et al. [31] reported synthesizing 1,3,4,5-tetrasubstituted pyrazole derivatives and their in vitro anti-inflammatory effect (see Figure 3). Compound **117a** showed excellent inhibition (93.80%) compared to the standard diclofenac sodium (90.21%) at a 1 mM concentration. El-Karim et al. [32] reported compounds **118a**–**118f** (edema inhibition% = 98.16%, 96.73%, 88.81%, 81.5%, 76.17%, and 76.68%, respectively) as potent candidates producing rapid onset and a long duration of anti-inflammatory activity, as well as a good safety GIT profile. Meanwhile, the analgesic evaluation revealed that **118b**–**118e** produced potent and long-acting analgesia accompanied by a significant inhibition of the inflammatory cytokine TNF-α level compared to the standard drugs. The inhibition in the protein denaturation of bovine albumin with IC_50_ of 34.1 μg/mL using diclofenac sodium as the standard drug (IC_50_ = 31.4 μg/mL). Out of the 15 novel compounds synthesized by Akhtar et al. [33], **123a**–**123d** demonstrated a significant in vitro anti-inflammatory activity, with IC_50_ values of 71.11, 81.77, 76.58, and 73.35 μg/mL, respectively, compared with the standard diclofenac. The benzylidene substituent attached remarkably influenced the anti-inflammatory potency. Abdellatif et al. [34] synthesized a new series of pyrazole derivatives. The inhibition efficacy of the target compounds to ovine COX-1 and human recombinant COX-2 was analyzed using an immune enzyme assay (EIA) kit. Most of the tested compounds showed high COX-2 inhibitory activity with *IC*_50_ values ranging from 0.02–0.04 μM. Meanwhile, **119a** and **119b** had the most suitable COX-2 selectivity index (SI = 462.91 and 334.25, respectively), superior to celecoxib (SI = 313.12) and indomethacin (SI = 1.37). Compounds **119a** and **119b** (SO_2_NH_2_ as the selective COX-2 pharmacophore) also showed the highest anti-inflammatory activity (ED_50_ = 136 and 126 μmol/kg, sequentially). In addition, they have the lowest ulcerogenic liability (Ulcer Index = 1.25 and 1.00, respectively), reflecting their expected safe GI profiles. Shi and coworkers [35] discovered **120** as the most potent anti-inflammatory agent (IC_50_ = 3.17 μM), with low toxicity and strong inhibitory NO release (inhibitory rates (IR) = 90.4% at 10 μM). This compound also showed potent inhibition of iNOS with an IC_50_ value of 1.12 μM. The treatment of compound **120** on acute inflammatory models in AA rats displayed a remarkable inhibitory effect on hind paw swelling and body weight loss, comparable to the effect identified in the aspirin-treated group. Of all the compounds investigated by Sivaramakarthikeyan et al. [36], the *para*-nitrophenyl moiety linked to a pyrazole conjugate **121** (93.53 ± 1.37%) displayed the highest anti-inflammatory activity in the anti-inflammatory assay using the protein denaturation method. This is superior to the standard, diclofenac sodium (90.13 ± 1.45%). Nayak et al. [37] revealed that compound **122a** showed remarkable sodium and celecoxib, which showed IC_50_ values of 55.65 and 44.81 μg/mL, respectively. The potent compounds were further evaluated for their in vitro COX-2 inhibitory activities using an enzyme immunoassay. Compound **123d** demonstrated able selectivity toward COX-2 with a selectivity index (SI) of 80.03 compared with the standard celecoxib, with an SI of 95.84. Dimmito et al. [38] reported that compound **124a** displayed a good analgesic effect after subcutaneous and intracerebroventricular management in vivo. Additionally, **124a** showed an excellent anti-inflammatory effect after subcutaneous administration, indicating prospective activity at the periphery. Harras and colleagues [39] synthesized a series of pyrazole derivatives and evaluated their in vitro COX-1/COX-2 inhibition and in vivo anti-inflammatory activity using the carrageenan rat paw edema model. It was noted that the targeted compounds exhibited more potent inhibitory activity against COX-2 than COX-1. Meanwhile, all compounds’ selectivity indexes (SI) were analyzed and compared to celecoxib (SI = 8.17). Compounds **125a** and **125b** displayed an outstanding COX-2 selectivity index of 8.22 and 9.31, respectively. Meanwhile, the histopathological investigation of the rats’ stomach, liver, and kidneys revealed that **125a** and **125b** triggered minimal degenerative changes, suggesting these derivatives’ safety. Sivaramakarthikeyan et al. [40] reported the anti-inflammatory activity of the pyrazole derivatives. The derivative, lacking substitution on the aryl entity **126**, exhibited the highest anti-inflammatory profile. Abdellatif et al. [41] synthesized a series of substituted pyrazole derivatives. The targeted compounds were screened for their COX-1/COX-2 inhibitory activity. Additionally, the carrageenan-induced rat paw edema model and histopathological study were demonstrated to examine their anti-inflammatory effectiveness and gastric safety. Compound **127** was the most potent anti-inflammatory agent (ED_50_ = 65.6 μmol/kg) compared to the reference drug, celecoxib (ED_50_ = 78.8 μmol/kg). In addition, the potent compound possessed minimum ulcerogenic (Ulcer Index = 7.25) Figure 3.

A new series of thiazolidindione **128** and thiazolidinone **129** containing a pyrazole core has been synthesized as hybrid structures [42]. The synthesized compounds were further evaluated for COX-1/COX-2 in vitro anti-inflammatory activity and ulcerogenic liability (Figure 3). The most COX-2-selective derivatives **128a** and **128b** and **129a** and **129b** showed the highest anti-inflammatory activities and the lowest ulcerogenic. Among the potent compounds, the thiazolidindione with a methoxy substituent **129b** displayed excellent activity against COX-2 (IC_50_ = 0.88 μM) with the highest COX-2 selectivity index (SI = 9.26). While compound **128c** with a methoxy substituent displayed the highest potent inhibitory against COX-2 (IC_50_ = 0.62 μM) with the highest COX-2 selectivity index (SI = 8.85). The highest anti-inflammatory (AI) activities were observed in **129a** and **128b** (after 1 h, AI = 82.34 and 81.15%; after 3 h, AI = 79.00 and 97.68%; and after 5 h, AI = 80.15 and 97.68%, respectively). Additionally, **1****28a** was slightly more potent (ED_50_ = 79.12 μmol/kg) than celecoxib (ED_50_ = 82.2 μmol/kg), while **128b** showed a superior ED_50_ value of 5.63 μmol/kg with a more than 14-fold effectiveness of celecoxib. Some pyrazolopyrimidine hybrids were prepared using Schiff base by Abdelall and coworkers [43]. All the synthesized compounds were evaluated in vivo against carrageenan-induced rat paw edema as anti-inflammatory agents. Regarding the anti-inflammatory activity compounds, **130** and **131** showed excellent activity compared to celecoxib. Thangarasu et al. [44] reported the anti-inflammatory effect of pyrazole moieties, and compound **132b** was found to have dominated activity potentials with an IC_50_ value of 3.5 nM in the COX-2 inhibition studies.

Murahari et al. [45] designed and synthesized novel pyrazole-based derivatives using the ligand-based approach. Among the synthesized compounds, **133** showed excellent in vivo anti-inflammatory activity with 0.8575 mmol/kg as ED_50_. The design and synthesis of novel thiophene–pyrazole hybrids have been investigated [46]. The thienopyrimidine analogs **134a**–**134b** and the thiophene derivative **135** are promising nontoxic, gastrointestinal-safe anti-inflammatory candidates with good oral bioavailability and physicochemical properties. A series of 1,2,3-triazole-linked 3-(1,3-diphenyl-1*H*-pyrazol-4-yl)acrylates was synthesized following a multi-step reaction [47] (Figure 4). Three of the evaluated compounds demonstrated significant anti-inflammatory activity, with IC_50_ values of 60.56, 57.24, and 69.15 μg/mL for compounds **136a**–**13c**, respectively, comparable to the standard diclofenac sodium with an IC_50_ value of 54.65 μg/mL.

Taher and colleagues [48] reported the synthesis and pharmacologic evaluation of novel pyrazole and pyrazoline derivatives. The study presents the effect of lengthening the carbon chain in different pyrazole derivatives bearing various amine moieties. Their results showed that lengthening of the aliphatic chain in **137a–137c** (26.19%, 30.95%, and 28.57%, respectively) led to higher activity. Meanwhile, the cyclization of chalcones into pyrazolines were more potently anti-inflammatory in compounds **138**, **139a** and **139b** (21.43%, 26.19%, and 28.57%. Compounds **138** and **140** exhibited the highest analgesic activity among all the examined compounds (75.9% and 84.5%, respectively). Mustafa et al. [49] presented a novel series of celecoxib derivatives. The in vivo anti-inflammatory activity of the synthesized compounds was evaluated using celecoxib as a reference standard by the paw oedema model on albino Wistars. Most of the compounds showed higher in vivo anti-inflammatory activity compared to celecoxib. Different substituents on the triazole moiety played a crucial role in the percentage inhibition of anti-inflammatory effects at 1h. Derivatives with chlorine atoms **141a**–**141d** and the nitro derivative **141e** showed good anti-inflammatory potency (Figure 4). A series of novel benzophenones conjugated with an oxadiazole sulfur bridge pyrazole has been designed, synthesized, and characterized [50]. It was afterward evaluated for anti-inflammatory and analgesic effects. Among the series, compound **142** (65.38% edema inhibition) with an electron-withdrawing group (fluoro) at the *para* position of the benzoyl ring of benzophenone was characterized by great activity compared to the standard drug. The analgesics activity data also revealed that compound **142** was the highest potent compound among the compounds evaluated for an analgesic effect on the acetic acid-induced writhing response and thermal pain (see Figure 4).

A novel series of pyrazole hybrids, such as pyrazole-thiohydantoin and pyrazole-methylsulfonyl, was synthesized by Abdellatif et al. [51]. The hybrids were evaluated in vivo for their anti-inflammatory activity (Figure 5). Compounds **143a–143d** were found to have the most active anti-inflammation. The unsubstituted **143b and 143d** showed comparable ED50 (78.90 and 88.28 μmol/kg) with celecoxib (ED_50_ = 78.53 μmol/kg), while the methoxy-substituted compounds **143a**, **143c**, and **143e** (ED_50_ = 62.61, 55.83, and 58.49 μmol/kg, respectively) showed superior activity to celecoxib. Thirteen pyrazole derivatives were synthesized and evaluated for their anti-inflammatory activity (in vitro and in vivo) and ulcerogenic liability [52]. Nine compounds **144**–**146** exhibited a moderate to high edema inhibition percentage (78.9–96%) than celecoxib (82.8%). Additionally, they were found to have potent COX-2 inhibitory activity, with the IC_50_ values ranging from 0.034 to 0.052 μM. Compound **145a** was the benign pyrazole with respect to the ulcerogenic effect (UI = 0.7) on the stomach, which may be ascribed to its high COX-2 enzyme selectivity (SI = 353.8), while compounds **144a**, **145b**, and **146a**–**c** exhibited ulcer index values (UI = 0.8–2) comparable to celecoxib. Sulphonyl derivatives **147** and **148** have been reported to be selective for the COX-2 isozyme with COX-2 selectivity indexes of 9.78, 8.57, 10.78, and 10.47, respectively, compared to celecoxib (S.I. = 8.68) [53]. Meanwhile **147** and **148** were observed as excellent anti-inflammatory derivatives with ED_50_ = 51.51, 46.98, 53.65, and 54.45 μmol/kg better than celecoxib (ED_50_ = 76.09 μmol/kg). Gedawy et al. [54] reported novel pyrazole sulfonamide derivatives as dual COX-2/5-LOX inhibitors. The benzothiophen-2-yl pyrazole carboxylic acid derivative **149** showed the most potent analgesic and anti-inflammatory activity superior to celecoxib and indomethacin. It showed potent COX-1, COX-2, and 5-LOX inhibitory activities, with IC_50_ of 5.40, 0.01, and 1.78 μM, respectively, showing a selectivity index of 344.56 superior to the reference standards (see Figure 5).

A new pyrazole sulfonate series has been reported [55]. Among the series, 4-iodophenyl 5-methyl-3-(p-tolyl)-1H-pyrazole-1-sulfonate **150a** and phenyl 5-methyl-3-(4-(trifluoromethyl) phenyl)-1H-pyrazole-1-sulfonate **150b** displayed superior anti-inflammatory activity (% inhibition of auricular edemas = 27.0 and 35.9, respectively); while the in vivo analgesic activity of **150c** and **150d** was more effective with an inhibition of 50.7% and 48.5% separately), and compounds **150a, 150c**, and **150d** were identified as selective COX-2 inhibitors (SI = 455, 10,497, and >189, respectively). In addition, the acute oral toxicity in vivo analysis showed lethal doses of 50 (LD_50_) of **150a** and **150d** to mice to be more than 2000 mg/kg. A novel series of 1,5-diaryl pyrazole-3-carboxamides was synthesized and evaluated against COX-1, COX-2, and sEH enzymes as dual COX-2/sEH inhibitors [56]. The anti-inflammatory activities of compounds **151a**–**c** were superior (edema inhibition percentages of 62%, 71%, and 65%) to the reference drug celecoxib (22%). Compounds **151a**–**c**, the most potent dual COX-2/sEH inhibitors in vitro, displayed the highest analgesic activity, with a % inhibition of 62.68, 71.64, and 67.16, respectively, and potencies 4.66, 5.33, and 5, respectively. Furthermore, compounds **151b** and **151c** substantially decreased the serum concentration of TNF-α with a % inhibition of 77% and 75%, respectively, when compared to celecoxib (64%). Compounds **152** and **153** have been reported as promising anti-inflammatory agents [57]. The compounds inhibited the lipoxygenase enzyme with IC_50_ values of 2.17 ± 0.12 and 2.53 ± 0.06 μM, respectively, compared to the standard quercetin (IC_50_ value = 3.35 ± 0.01 µM) (see Figure 5).

### 4.2. Anticancer

Compound 5-(5-Bromo-1-methyl-1H-indol-3-yl)-1-(4-cyano-phenyl)-3-methylsulfanyl-1H-pyrazole-4-carbonitrile **117b** showed significant cytotoxicity against MCF 7 (GI_50_ = 15.6 µM) with low cytotoxicity against a normal Vero cell line [31] (see Figure 4). Sivaramakarthikeyan et al. [36] reported that the evaluation of the anticancer potency of the synthesized pyrazole-benzimidazole hybrids revealed that the hybrids bearing a *para*-fluorophenyl unit tethered at the pyrazole nucleus (**121b**) showed the highest activity against both the pancreatic cancer cells (SW1990 and AsPCl) with IC_50_ of 30.9 ± 0.77 and 32.8 ± 3.44 µM compared to the reference compound, gemcitabine 35.09 ± 1.78 and 39.27 ± 4.44 µM. Akhtar et al. [33] revealed that **123b** was active against A549, SiHa, COLO205, and HepG2 cancer cell lines, with IC_50_ values of 4.94, 4.54, 4.86, and 2.09 μM. The potent compound **123b** was also nontoxic against normal cells (cell line HaCaT), with an IC_50_ value greater than 50 μM (see Figure 6).

Sivaramakarthikeyan et al. [40] showed that **126** exhibited significant activity against both the pancreatic cell lines—namely, AsPC1 and SW1990—with IC_50_ values of 30.3 ± 0.45, 32.4 ± 0.65 µM and noncancerous cell—namely, MRC5 with an IC_50_ value of 55.5 ± 3.50 µM (Figure 3). The synthesis and anticancer evaluation of a series of 1,2,3-triazole linked 3-(1,3-diphenyl-1*H*-pyrazol-4-yl)acrylates have been carried out [47]. Compound **136b** showed the most promising anticancer effects among the synthesized compounds, with IC_50_ values of 1.962, 3.597, 1.764, and 4.496 μM against the A549, HCT-116, MCF-7, and HT-29 cell lines, respectively. The sulphamoyl derivatives **148a** and **148b** exhibited the most potent activity, with IC_50_ of 5.34 and 6.48 μM against A549 and IC_50_ of 4.71 and 5.33 μM against MCF-7. Additionally, the compounds showed potent activity, with IC_50_ values of 4.39 and 5.12 μM against HCT-116 and IC_50_ of 3.66 and 4.37 μM against PC-3 [53]. A further analysis disclosed that compounds **148a** and **148b** arrested the cell cycle activity on the PC-3 cell line with greater selectivity. Meanwhile, the antiproliferation potency of compounds **148a** and **148b** on PC-3 cells is due to cell cycle arrest and apoptosis-inducing activity categorized by the Bax/Bcl-2 ratio increase (see Figure 7).

Compounds **154**–**156** were effective VEGFR-2 kinase inhibitors with IC_50_ of 913.51, 225.17, and 828.23 nM, respectively, compared to sorafenib (IC_50_ = 186.54 nM) [58]. Further, the cellular mechanistic studies of **156** revealed its promptness towards pre-G1 apoptosis and cell growth termination at the G2/M phase.

A new series of pyrazolo [1,5-*a*]pyrimidine derivatives has been designed and evaluated for their cytotoxic activities on a human breast adenocarcinoma cell line (MCF-7) and colon cancer cell line (HTC-116) [59]. The results revealed that **157** was the most potent among the tested compounds against HTC-116, with an IC_50_ value of 1.51 μM, while **158** (IC_50_ = 7.68 μM) displayed an excellent cytotoxic effect superior to reference doxorubicin against MCF-7. The tetrahydrothiochromeno [4,3-c]pyrazole derivatives were synthesized and evaluated for anticancer activity using MTT [60]. Most of these compounds showed potential anticancer activity and low cytotoxicity on the normal cells in vitro. Compounds **159a** and **159b** displayed excellent anticancer activity, with IC_50_ values of 15.43 μM and 20.54 μM towards MGC-803, respectively. Additionally, the potent compounds **159a** and **159b** induced G2/M cell cycle arrest and apoptosis in MGC-803 cells. New pyrazole Schiff bases containing azo groups **160** have been reported as promising anticancer agents [61]. A new series of novel pyrazole-containing imide derivatives were synthesized and evaluated for their anticancer activities against the A-549, Bel7402, and HCT-8 cell lines [62]. Among the evaluated compounds, **161a**–**161d** exhibited potent inhibitory activity against the A-549 cell line, with IC_50_ values at 4.91, 3.22, 27.43, and 18.14 μM, respectively, superior to 5-fluorouracil (IC_50_ = 59.27 μM). Additionally, **161a**–**161c** exhibited substantial inhibitory activity towards the HCT-8 and Bel7402 cell lines (see Figure 7).

The cytotoxic activity of spirocycloadducts and *N*-arylpyrazole hybrids against the HeLa cancer cell line was evaluated using an MTT assay [63]. Spiro[indenoquinoxaline-pyrrolizidine]-*N*-arylpyrazole conjugate **162** bearing a *p*-chlorophenyl substituent exhibited the highest antiproliferative activity against cancer cell line HeLa (IC_50_ = 1.93 μM). The IC_50_ is comparable to camptothecin’s standard drug (IC_50_ = 1.66 μM) (see Figure 7). A novel series of 1,2,3-triazole-pyrazole hybrids were designed and synthesized using the Cu-catalyst [64]. The synthesized compounds were evaluated for anticancer activity using three cancer cell line panels. Compound **163** was the most potent cytotoxic candidate for HepG-2, HCT-116, and MCF-7, with IC_50_ = 12.22, 14.16, and 14.64 µM, respectively, comparable to the standard drug doxorubicin (IC_50_ = 11.21, 12.46, and 13.45 µM). Ragab et al. [65] revealed compounds **164** and **165** as promising leads for colon cancer treatment. Compounds **164** and **165** were active against the KM12 cell line, with an IC_50_ value of 1.73 and 1.21 μM and high selectivity index (SI) (18.82 and 35.49, respectively). Compared to the standard drug 5-FU with an IC_50_ value of 12.26 μM and SI value of 1.93. The potent compound displayed selective cytotoxic activity against KM12 cells in the annexin V-FITC staining assay.

Mohamady et al. [66] designed and synthesized diarylpyrazole derivatives. The compounds were screened against the MCF7 and HepG2 cell lines. Among the evaluated compounds, **166**, which contained a thiophene ring, was observed to have the highest antiproliferative activity against HepG2 cells, with an IC_50_ of 0.083 μM. The compound caused cell cycle arrest at the G2, and the 7.7-fold increase in caspase-3 confirmed its apoptotic effect on HepG2 cells. Additionally, **166** caused a noticeable decrease in Hsp90 proteins (Akt, c-Met, c-Raf, and EGFR) and a 1.57-fold upsurge in Hsp70.

Pyrazolothiazole-substituted pyridine conjugates were synthesized and evaluated for cytotoxicity activity [67]. Compound **167**—namely, 4-amino-7-(2-(1,5-dimethyl-1*H*pyrazol-3-yl)-4-methylthiazol-5-yl)-2-oxo-5-(thiophen-2-yl)-1,2-dihydro-1,8-naphthyridine- 3-carbonitrile—has the highest cytotoxicity activity towards PC-3 (IC_50_ = 17.50 µM), NCI-H460 (IC_50_ = 15.42 µM) and Hela (IC_50_ = 14.62 µM), comparable to the anticancer potential of standard drug etoposide (IC_50_ = 17.15, 14.28, and 13.34 µM, respectively).

Wang et al. [68] investigated a new series of pyrazole-naphthalene derivatives. The synthesized compounds were evaluated for their anticancer activity against breast cancer cell lines (MCF-7). Compound **168** (IC_50_ = 2.78 ± 0.24 μM), with substituted ethoxy at position 4 of the phenyl ring, exhibited the highest activity, and the activity was 5-fold more active than the reference drug cisplatin (IC_50_ = 15.24 ± 1.27 μM. Additionally, **168** showed inhibited tubulin polymerization with an *IC*_50_ value of 4.6 μM. Mohamed and coworkers [69] recommended cyanoacrylamide compound **169** as a new promising chemotherapeutic agent. The compound exhibited high cytotoxic activity toward colorectal carcinoma. Compounds **170** and **171** displayed excellent antiproliferative activities towards HDAC2, with IC_50_ values of 0.25 and 0.24 nM, respectively, and CDK2 with IC_50_ values of 0.30 and 0.56 nM, respectively. The potent compounds **170** and **171** pointedly inhibited the movement of the A375 and H460 cells, arrested the cell cycle in the G2/M phase, and promoted apoptosis in A375, HCT116, H460, and Hela cells, related to proliferating the intracellular reactive oxygen species (ROS) levels. Notably, **170** has good pharmacokinetic properties, with an intraperitoneal bioavailability of 63.6% in ICR mice. Additionally, the compound was effective in vivo for antitumor activity in the HCT116 xenograft model. Thus, the authors proposed compound **171** as a favorable agent for treating malignant tumors. Burgart and colleagues [70] found 4-aminopyrazole derivatives **172a** and **172b** to be cytotoxic against HeLa cells and human dermal fibroblasts cancer cells. A novel series of pyrazole-arylacetamide hybrids has been synthesized and evaluated for cytotoxicity activity. Compounds **173a** and **173b** exhibited a potent cytotoxic effect on the MCF-7 cancer cell line, with *IC*_50_ values of 0.604 μM and 0.665 μM compared to the standard drug cisplatin (0.636 ± 0.458 μM) (see Figure 7).

Answer et al. [71] synthesized some novel pyrazole hybrids using 5-amino-3-(4-(dimethylamino) phenyl)-1-phenyl-1*H*-pyrazole-4-carbonitrile as a precursor. Including different nucleophilic and electrophilic compounds, among the synthesized compounds, the anticancer activities of the synthesized compounds **174**, **175**, **176**, and **177** (6.50 ± 0.5, 3.74 ± 0.3, 3.18 ± 0.2, and 8.67 ± 0.9 µM, respectively) exhibited strong cytotoxic activity against MCF-7 and HCT-116 (7.80 ± 0.6, 4.93 ± 0.3, 4.63 ± 0.4, and 10.02 ± 1.0 µM, respectively) (see Figure 7). Hassan et al. [72] synthesized a novel series of pyrazolopyrimidines and screened the compounds against a panel of 60 human cancer cell lines. Compounds, 5-amino-1H-pyrazole-4-carbonitrile derivative **177**, pyrazolo [5,1-b]quinazoline-11-carbonitrile derivative **178**, and 1-amino-2,4-dihydro-5*H*-benzo [4,5]imidazo [1,2 c]pyrazolo [4,3-e]pyrimidin-5-one **179** exhibited anticancer activity against some cancer cell lines (Figure 7). Compounds **180a**–**180c** have been reported to displayed anticancer [73] compounds **180a**–**180c** bearing an electron-donating group, such as methoxy substituent at the *para* position; the 3,4 dimethoxy and 3,4,5 trimethoxy derivatives demonstrated noticeable cytotoxic activity with IC_50_ values of 0.604 μM, 1.057 μM, and 0.665 μM respectively, against the MCF-7 cancer cell lines.

Thiazolyl pyrazole carbaldehyde hybrids have been synthesized and screened for their in vitro anticancer activity by Mamidala and colleagues [74]. Compound **181** exhibited the highest antiproliferative activity against the HeLa, MCF-7, and A549 cancer cell lines, with IC_50_ values of 9.05 ± 0.04, 7.12 ± 0.04, and 6.34 ± 0.06 µM, respectively. Raghu et al. [75] designed and synthesized a new series of 1,3,5-triazine-based pyrazole hybrids with anticancer activity targeting the epidermal growth factor (EGFR) tyrosine kinase. Compounds **182a**–**182c** exhibited potent anticancer activity against the MCF-7 (human breast), HepG2 (human liver), HCT116 (human colorectal), PC-3 (human prostate), LoVo (human colon), and LoVo/DX (doxorubicin-resistant) cancer cell lines. According to the EGFR tyrosine kinase test, **182a**–**182c** demonstrated excellent activity, with an IC_50_ value of 395.1, 286.9, and 229.4 nM. Compared to the reference doxorubicin (63.8 nM) and erlotinib (103.8 nM), compound **182c**, with a trifluoromethyl group at the *para* position on the phenyl rings, exhibited the most potent anticancer activity. Thus, the anticancer activity of the tested compounds was affected by the physicochemical properties of the substituent on the pyrazole-bound phenyl nucleus. Compounds **183a–183c** have been reported as promising antiproliferative agents [76]. Suryanarayana et al. [77] synthesized a novel series of dinitrophenylpyrazole-bearing triazole and further investigated their anticancer activity using three tumor cell lines—namely, MCF-7, HeLa, and HeLa Caco-2. Among the synthesized compounds **184a**–**184c** with the methoxy group on the phenyl ring at the *ortho, meta*, or *para* position exhibited excellent inhibitory activity against the HeLa (*IC*_50_ = 4.0 μM, 5.0 μM, and 6.0 μM) compared to the standard drug combretastatin-A4 (IC_50_ = 9.0 μM). Compound **185c** exhibited excellent inhibitory activity against the MCF-7 cell line, with an IC_50_ value of 8.0 μM. Compound **185** has been reported as a promising anticancer agent that reduced the level of CDK2, stopped MCF-7 cells in the G0/G1 phase, caused ROS growth, damaged the MMP, and accelerated the apoptosis of MCF-7 cells [78] (see Figure 8). Among the new library of pyrazole derivatives investigated for antiproliferative activity by Signorello and coworkers, compounds **186a** and **186b** exhibited antiproliferative effectiveness [79]. Compound **186a** displayed moderate inhibition (25–30% at 10 μM) toward melanoma (SK-MEL-5, UACC-62) and renal cancer cell lines (UO-31 cancer cell lines). Compound **186b** displayed a broad spectrum of action (25–30%) towards leukemia (CCFR-CEM and RPMI-8226), non-small cell lung (NCI-H522), CNS (SF-295 and SNB-75), ovarian (OVCAR-4) and the breast (BT-549 and MDA-MB-468) cancer cell line. Additionally, compound **186b** demonstrated 50–60% inhibition towards melanoma (SK-MEL-5 and UACC-62), renal (CAK-1 and UO-31), and prostate (PC-3) cancer cell lines. The water-soluble, BBB4-loaded NPs (BBB4-G4K NPs, **187**), achieved from BBB4 in a non-bioactive, polyester-based, lysine-containing fourth-generation cationic dendrimer (G4K) has been reported to have an excellent antibacterial profile and highly selective toward the *Staphylococcus* genus [80]. The synthesis and biological screening of 5- pyrazolyl urea as potential antiangiogenic compounds were investigated by Morretta et al. [81]. Among the targeted compounds, compound **188a** displayed 100% inhibition on leukemia cancer cell lines, while **188b** and **188c** impeded non-small cell lung cancer cell lines. Compound **188d**, similar to the STIRUR-41 pharmacophore, exhibited 40% inhibition in the non-small cell lung cancer cell lines. Additionally, compounds **188e** and **188f**, containing a trifluoromethyl substituent on the urea moiety, displayed excellent inhibitory activity. Meanwhile, the mechanism of action detailed that **188e** may likely exert its antiproliferative activity by targeting different signaling pathways, including ERK/MAPK and phosphatases, or the crosstalk between these two associated intracellular mechanisms. Compound **188e** can regulate ERK1/2 phosphorylation and PP1g action.

### 4.3. Antibacterial

Nayak et al. [37] reported **122a**–**122g** as promising antibacterial agents (Figure 3). Ebenezer et al. [82] reported a designed and synthesized library of novel pyrazole–imidazo [1,2-α]pyridine scaffolds through a one-pot three-component tandem reaction. All selected compounds (zone of inhibition ˃9 mm) showed excellent bactericidal activity. In most cases, except for MRSA, the activity of the compounds was better than that of the standard ciprofloxacin. Compounds **189a**–**189e** had excellent activity against *S. aureus, E. coli*, *S. typhimurium*, *K. pneumoniae*, and *P. aeruginosa*, a with minimum bactericidal concentrations (MBC) <0.1 μg/mL. Pyrazoloquinoline derivatives have been synthesized and their antibacterial activity evaluated using the Agar diffusion method [83]. The ketonic compounds **190a** and **190b** showed activity percentages of 112% and 95% (*Streptococcus pneumoniae*) and 86% and 83%, respectively (*Bacillus subtilis*), compared to the standard control. The halogenated ketonic derivative **190b** showed improved activity compared to gentamicin (109%), and no activity was observed in *Pseudomonas aeruginosa*. Hansa and coworkers [84] reported 4-4-(anilinomethyl)-3-[4-(trifluoromethyl)phenyl]-1H-pyrazol-1-ylbenzoic acid derivatives as potent anti-Gram-positive bacterial agents. Many of the evaluated compounds are potent growth inhibitors of Gram-positive bacteria and showed low toxicity of human cultured cells. Among the compounds, **191a** and **191b** exhibited excellent inhibition against *Staphylococcus aureus*. A novel series of multi-substituted benzo-indole pyrazole derivatives with antibacterial activity targeting DNA gyrase has been investigated [85]. Compound **192** exhibited excellent antibacterial activity against four drug-resistant *E. coli* bacteria strains (*E564*c, *E68*d, *E48*e, and *E109*, respectively) with IC_50_ values of 7.0 and 17.0, 13.5, and 1.0 μM, respectively. The substitution of fluorine or chlorine at R_2_ enhanced the bacteriostatic effect. The derivative bearing a Cl atom at R_1_ and R_2_ exhibited a superior antibacterial effect. Additionally, compound **192** displayed potent inhibition against DNA gyrase, with IC_50_ values of 0.10 μM in the in vitro enzyme inhibitory assay. A library of twenty-three novel pyrazole–phenylthiazole hybrids was synthesized and screened for antimicrobial activity against five bacterial species and two fungi [86]. Compound **193** displayed a promising antibacterial effect against the Gram-positive methicillin-resistant *Staphylococcus aureus* (MRSA) strain with a MIC value of 4 μg/mL. Compound **193** was nontoxic to mammalian cells—namely, human embryonic kidney cells and human red blood cells. All the synthesized compounds except **194** (moderate growth inhibition of 40.8%) showed poor antifungal activity. Analogs of pyrazole–thiazolidinone and pyrazole–thiosemicarbazone were designed using a molecular hybridization approach and further synthesized through a Vilsmeier–Haack approach [87]. The compounds were tested for antimicrobial activity against two Gram-positive bacteria, such as *Staphylococcus aureus* and methicillin-resistant *Staphylococcus aureus*. Additionally, four Gram-negative bacteria such as *Escherichia coli*, *Salmonella typhimurium*, *Klebsiella pneumonia,* and *Pseudomonas aeruginosa* were used in the biological assay. Derivatives **195** and **196** appeared as the most active antimicrobial compounds, with an MBC value of <0.2 μM against MRSA and *S. aureus*. The presence of 2,4-dichloro group on **195** enhanced its antibacterial activity. A new pyrazole containing isonicotinoyl derivatives from substituted chalcones and isoniazid by using sulfamic acid and their pharmacological activity evaluation have been investigated [88]. All examined compounds showed inferior activity against *E. coli*. The MIC values divulged that compounds **197a**, **197c**, and **197d** (MIC values = 14, 17, and 14 µM) exhibited good antimicrobial activity against *Staphylococcus aureus*, while compounds **197a** and **197b** (MIC values = 14 and 29 µM) displayed superior antimicrobial activity against *Pseudomonas aeruginosa*. Compounds **197b** and **197d** (MIC values = 117 and 114 µM) exhibited noticeable activity against *Salmonella typhi*. Notably, the electro-donating group at the R position improved the antimicrobial activities more than the insertion of the electro-withdrawing group. Additionally, electron-donating substituents at position three enhanced the antimicrobial activity compared to position four. The one-pot reaction of *bis*-hydrazonoyl bromide with active methylene reagents furnished new *bis*-thiazolyl-pyrazole derivatives [89]. The insertion of acetyl (COCH_3_) and methyl (CH_3_) groups improved the activity of compound **198** against Gram-positive strains with MIC values 2, 8, and 8 μM towards *S. aureus*, *B. subtilis,* and *E. faecalis,* respectively. In comparison, **198** showed moderate antifungal and no substantial activity against Gram-negative strains with MIC *>* 32 μM. The most potent compound against the Gram-positive bacterial strains was **200** with MIC values of 0.12, 1, and 0.5 μM for *S. aureus*, *B. subtilis*, and *E. faecalis,* respectively, compared to the standard drug vancomycin (1 to 2 μM). The lipophilic aryl substituent at position five and cyano at position four in the pyrazole ring improved the antibacterial activity of **200**. Compounds **198**, **199b**, and **200** exhibited more potent inhibitory activity of DHFR with IC_50_ values (6.34 ± 0.26, 7.49 ± 0.28, and 3.81 ± 0.16 μM), respectively, compared with Trimethoprim (8.34 ± 0.11 μM). However, *bis*-1-(thiazol-2-yl)-5-(amino)-1*H*-pyrazole-4-carbonitrile derivative **199a** was revealed to be the least inhibited toward DHFR in comparison to Trimethoprim and other tested derivatives, with an IC_50_ value 19.38 ± 0.68 μM, and that may be related to the presence of the carboxamide group in position four at the pyrazole ring rather than acetyl or carbonitrile as pyrazole derivatives **198**, **199b**, and **200.** Desai and colleagues [90] synthesized analogs of pyrazole, pyrazoline-clubbed pyridine compounds, and examined their antibacterial and antifungal activities. Among the test compounds, **201a** (3-OH) and **201b** (4-F) exhibited good activity (MIC = 50 μg/mL) against *S. aureus* and *E. coli*, respectively. Compound **201c** (2,4-dichloro) showed superior activity (MIC = 12.5 μg/mL) against *P. aeruginosa* and very good activity (MIC = 25 μg/mL) against *S. pyogenes* compared to the standard drug ampicillin (100 μg/mL) and chloramphenicol (50 μg/mL). The derivatives bearing electron-donating groups (2-OH, 3-OH, 4-CH_3_, and 4-OH-3-OCH_3_) showed significant antifungal and antibacterial activity, while the derivatives bearing electron-withdrawing groups (4-F and 2,4-dichloro) showed an augmentation in the antibacterial potency. Compound **202** has been reported as a promising antibacterial agent (*B. cereus, S. aureus, P. aeruginosa*, and *E. coli*) with the closest inhibition zones (17–20 mm) [91] (see Figure 9).

### 4.4. Antifungal

Compounds **201d** (4-CH_3_) and **201e** (4-OH-3-OCH_3_) showed significant antifungal activity towards diverse fungal strains [90]. Compound **201d** (4-CH_3_) displayed significant activity (MIC = 12.5 μg/mL) towards *A. niger,* and compound **201e** (4-OH-3-OCH_3_) showed excellent activity (MIC = 12.5 μg/mL) against *C. albicans* and *A. clavatus* (Figure 9). Othman et al. [92] reported novel heterocyclic hybrids of pyrazole and their antimicrobial activity. Compounds bearing a benzenesulphonamide group fused with 3-methyl-5-oxo-1H-pyrazol-4-(5*H*)-ylidene) hydrazine **203** and 6-amino-7-cyano-3-methyl-5*H*-pyrazolo [4,3-c]pyridazine **204** showed significant and broad-spectrum antimicrobial activity. Compound **204** displayed excellent antifungal activity toward *A. fumigates* with a MIC value of 0.98 µg/mL. While **203** and **205** are equipotent with the reference, amphotericin B against *A. fumigates* with a MIC value of 1.95 µg/mL exhibited 2-fold decrease in the effectiveness compared to the standard, ciprofloxacin against *S. pneumonia* (MIC = 1.95 and 0.98 µg/mL, respectively). Compound **203** was equipotent in the antifungal activity with the reference (MIC = 3.9 µg/mL) against *C. albicans*.

A series of novel pyrazole-thiazole carboxamides were designed, synthesized, and investigated for their antifungal activity [93]. The outcomes showed that compounds **206**, **207**, and **208** have promising in vitro activities against *Rhizoctonia cerealis*, with EC_50_ values from 1.1 to 4.9 mg/L, superior to thifluzamide (EC_50_ = 23.1 mg/L). The antifungal activity of **207** (EC_50_ value = 1.1 mg/L was ~21-fold more active than thifluzamide and ~ 2-fold more active than compound **206** (EC_50_ = 2.0 mg/L). Meanwhile, **208** exhibited excellent antifungal activity towards *S. sclerotiorum*, with an EC_50_ value of 0.8 mg/L, which was ~6-fold higher than thifluzamide (EC_50_ = 4.9 mg/L). The conjugates bearing an aniline moiety with a single substituent at the *ortho*- or *meta*-position exhibited promising antifungal activity. Additionally, the in vivo antifungal assay showed that **206** (90% at 10 mg/L) exhibited higher antifungal activity than thifluzamide against *R. solani* (90% at 10 mg/L). The synthetic route to pyrazole-4-formylhydrazine derivatives bearing a diphenyl ether fragment was reported by Wang et al. [94]. The synthesized compounds were evaluated for their antifungal activity by targeting a succinate dehydrogenase. Among the tested compounds, **209a** against *Rhizoctonia solani,*
**209b** against *Fusarium graminearum*, and **209c** against *Botrytis cinerea*, exhibited superior antifungal activity. The compounds displayed EC_50_ values of 0.14, 0.27, and 0.52 μg/mL higher than carbendazim against *R. solani* (0.34 μg/mL) and *F. graminearum* (0.57 μg/mL), along with penthiopyrad against *B. cinerea* (0.83 μg/mL). Compound **209a** was ~2- and 15-fold higher than the marketed fungicides carbendazim (0.34 μg/mL) and boscalid (2.21 μg/mL) towards *R. solani*. The results from the determination of the inhibitory effects of **209a** against the SDH collected from the mycelia of *R. solani* showed *IC*_50_ values of 3.99 μM (1.58 μg/mL). The in vivo anti-*R*. *solani* control effectiveness of the most potent compound, **209a** (73.25% at 200 μg/mL), was significantly superior to carbendazim under the same conditions (59.81%). Compound **210** has been reported as an excellent antifungal agent with equivalent activity to the marketed fungicide drug thifluzamide, and its *EC*_50_ value was 0.022 mg/L against *R. solani* [95] (see Figure 10).

A synthetic route to substituted 3-(trifluoromethyl)-4,5-dihydro-1*H*-furo [2,3-c] pyrazole conjugates using the [3 + 2] Michael/Alkylation approach was developed by Tan et al. [96]. The antifungal activity of the synthesized compounds was further examined, and **211a** exhibited excellent antifungal activity against *A. solani* with IC_50_ values of 5.44 μM. Compounds **211b** and **212** also displayed good antifungal effects, with corresponding IC_50_ values of 9.00 and 31.45 μM, respectively. The antifungal effects of the most active compounds **211a** and **211b** and **212** were relatively superior to the standard compound cycloheximide (IC_50_ = 71.00 μM). The conjugates bearing the electron-deficient group on the aromatic ring displayed higher inhibitory effects. At the same time, the compound bearing an electron-donating substituent on the aromatic ring showed reduced inhibitory effects. Compounds **213, 214a** and **214b**, and **215** have been unveiled to exhibit good antifungal activity toward *Fusarium Oxysporum f. sp.* and *Albedinis (FOA) fungus* with IC_50_ values ranging from 25.6 to 33.2 μg/mL [97]. Compound **216** has been reported to have superior antifungal inhibitory activity against *C. albicans* (MIC= ≤146 μg/mL) compared to **217**, which displayed a MIC= ≤183 μg/mL. Meanwhile, **218a** and **218b** exhibited significant antifungal effectiveness concentrations against *C. albicans* compared to the reference compound cycloheximide with the corresponding MIC values ≥ 168 μg/mL and ≥165 μg/mL, respectively [98]. Piperazine-pyrazole-4-carboxylic acids have shown good antifungal inhibitory effects. Compounds **219a–219e** showed equipotent antifungal activity with the reference miconazole against *C. albicans* (MIC value = 78.1 μg/mL) [99] (see Figure 10).

Dong and coworkers [100] synthesized a series of novel pyrazole–4-carboxamides hybrids and further evaluated their antifungal activity (see Figure 10). Compound **220** was the most potent compound against *A. solani* in vitro, with an EC_50_ value of 3.06 μg/mL. It displayed 100% (10 μg/mL) inhibitory activity against *A. solani* in vivo compared to the standard drug boscalid. Makhanya et al. [101] revealed compounds **221a** and **221b** as promising antifungal agents. The antifungal assay showed that **221a** and **221b** exhibited significant inhibitory activity against *Saccharomyces cerevisiae* (zone inhibition (ZI) = 23 and 20 mm, respectively), with a MIC value of 0.18 µM. Comparable with the standard drug amphotericin, B. Bayazeed et al. [102] detected four chromen-3-yl-pyrazole derivatives: **222a**, **222b**, **223a**, and **223b** to have superior antifungal activity. Compared to the standard drug ketoconazole against *Aspergillus fumigatus* with the corresponding % ZI values of 164%, 147.1%, 158.8%, and 147.1%, respectively. In addition, compounds **222b** (% ZI value = 150%) and **223b** (% ZI value = 150%) have 1.5-fold superior activity against *C. albicans* compared to the reference ketoconazole (% ZI value = 150%). Notably, the insertion of two ester groups in compound **222a** improved its antifungal activity. Meanwhile, incorporating electron-donating groups (OCH_3_ and CH_3_) at the *para*-position of the aromatic ring in **222b**, **223a**, and **223b** enhanced their antifungal activity. Wang et al. [103] reported a novel series of pyrazole-4-acetohydrazide derivatives targeting fungal SDH, further evaluating their antifungal properties towards *R. solani*, *F. graminearum*, and *B. cinerea*. Among the evaluated compounds, the antifungal activity of **224a** against *R. solani*, **224b** against *F. graminearum*, and **224c** against *B. cinerea* had EC_50_ values of 0.27, 1.94, and 1.93 μg/mL, respectively. These values were superior to the standard reference boscalid against *R. solani* (0.94 μg/mL) and fluopyram against *F. graminearum* (9.37 μg/mL) and *B. cinerea* (1.94 μg/mL). Additionally, the compounds with hydroxyl group substituents at the R_1_ position displayed higher anti-*R. solani* activity than the corresponding conjugates bearing an ethoxy group substituent. The in vivo studies detailed that compound **224a** was effective toward *R. solani* (79.83% at 200 μg/mL), comparable to validamycin (86.56%) and thifluzamide (83.49%). Compound **224a** was predicted as an SDH inhibitor (see Figure 11).

A series of new pyrazole-4-carboxamide conjugates were designed and synthesized by Wu et al. [104]. The synthesized compounds were evaluated for their antifungal activity using four phytopathogenic fungi (*G. zeae*, *F. oxysporum*, *C. mandshurica*, and *P. infestans*). The EC_50_ values were 1.8 μg/mL for **225a** against *G. zeae*, 1.5 and 3.6 μg/mL for **225b** against *F. oxysporium* and *C. mandshurica*, respectively, and 6.8 μg/mL for **225c** against *P. infestans*. Meanwhile, the SDH enzymatic effectiveness unveiled corresponding IC_50_ values of 6.9, 12.5, 135.3, and 223.9 μg/mL, for **225c**, **222d, 225e**, and penthiopyrad, respectively. Incorporating substituents (CH_3_, F, or Cl) into the 2-phenyl and methyl into 2-pyridinyl positions enhanced the antifungal activity. While the introduction substituent into the 3-phenyl and 3-pyridinyl positions decreased the antifungal properties. Xia et al. [105] reported novel pyrazole carboxylate derivatives bearing thiazole as potent fungicides. The antifungal studies revealed compound **226** displayed superior activities against *Botrytis cinerea* and *Sclerotinia sclerotiorum*, with EC_50_ values of 0.40 and 3.54 mg/L, respectively. Compound **227** displayed superb antifungal activity against *Valsa mali*, with an EC_50_ value of 0.32 mg/L. The in vivo fungicide control studies against *B. cinerea* and *V. mali* revealed that compounds **226** and **228** at 25 mg/L, respectively, were influential on cherry tomatoes and apple branches. Compound **227** displayed an inhibitory activity toward SDH, with an IC_50_ value of 82.26 μM. Nevertheless, compounds **226** and **228** lack inhibitory activity toward SDH in the in vivo studies (see Figure 11).

### 4.5. Antidiabetics

Thiazolidindione **128a** and **128b** and thiazolidinone derivatives **129a** and **129b** displayed significant inhibitory activities against *α*- and β-glucosidase (% inhibitory activity = 62.15, 55.30, 65.37, and 59.08 for α-glucosidase and 57.42, 60.07, 58.19, and 66.90 for β- glucosidase, respectively) than the reference compounds: acarbose with % inhibitory activity = 49.50 for α-glucosidase and *D*-saccharic acid 1,4-lactone monohydrate with % inhibitory activity = 53.42 for β-glucosidase. Compared to pioglitazone and rosiglitazone, the potent compounds showed good PPAR-γ activation and hypoglycemic effectiveness [42]. Kattimani et al. [106] reported the ring alteration of 3-arylsydnone into 1-aryl-1*H*-pyrazole-3-carbonitriles via a [3 + 2] cycloaddition reaction and were subsequently converted into 5-(1-aryl-1*H*-pyrazol- 3-yl)-1*H*-tetrazole. The synthesized compounds were screened for in vivo antihyperglycemic activity using albino Wistar rats. Compounds **229a** and **229b** and **230a–230d** pointedly reduced the blood glucose levels and prevented vascular difficulties in streptozotocin-induced diabetic rats (Figure 12). Compounds **231** and **232** were potent inhibitors of the α-amylase enzyme [107]. Compound **231** showed excellent activity against α-amylase, with an IC_50_ value of 4.08 μg/mL, followed by **232** with an IC_50_ value of 7.59 μg/mL. The potency of the compounds was superior to acarbose (*IC*_50_ value = 8.0 μg/mL). Pogaku et al. [108] designed and synthesized new pyrazole–triazolopyrimidine hybrids as potent α-glucosidase inhibitors using a one-pot multicomponent approach. Among the evaluated compounds, **233a–c** bearing an electron-withdrawing group on the phenyl ring displayed significant inhibitory activity against the α-glucosidase enzyme. Compound **233a** with the 4-Cl group showed the highest inhibition, with an IC_50_ value of 12.45 μM, equipotent to the standard drug acarbose (IC_50_ value = 12.68 μM). In contrast, **233b** with a fluoro substitution at the *para* position displayed an IC_50_ value of 14.47 μM, followed by **233c** bearing a 4-NO_2_ group (*IC*_50_ value 17.27 μM). Karrouchi et al. [109] designed and synthesized a pyrazole-3-carbohydrazide, **234.** The in vitro α-glucosidase inhibition study of **234** showed good activity for a concentration of 0.08 mM with a percent inhibition of 79.83%, superior to acarbose (29%). The β- galactosidase evaluation displayed a good inhibitory activity with a percentage of 64.6%, comparable to quercetin (68%) for a concentration of 3.30 mM, while the α-amylase inhibition results revealed an inhibitory activity of 20.51% comparative to the acarbose with a percentage of 36% for a concentration of 3.53 mM. The Rhodanine–pyrazole conjugates were designed and synthesized by Singh et al. [110]. The compounds were further tested for their antidiabetic activity. Among the evaluated compounds, **235a** (IC_50_ = 2.259 × 10^−6^ mol/L) was the most potent compound, 42-fold superior to acarbose. The unsubstituted hybrid **235b** (IC_50_ = 2.854 × 10^−5^ mol/L) was 3-fold superior to acarbose. Meanwhile, **235c** (IC_50_ = 6.377 × 10^−5^ mol/L) and **235d** (IC_50_ = 1.325 × 10^−4^ mol/L) exhibited strong inhibitory activity against α-amylase comparable to acarbose. Compound **236** has been reported to showed a promising bifunctional antidiabetic effect [111]. A series of new benzo[*d*][1,2,3]triazol-1-yl-pyrazole-bearing dihydro-[1,2,4] triazolo [4,3-*a*]pyrimidine groups have been designed and synthesized [112]. All synthesized compounds were screened in vitro for α-glucosidase inhibition, anticancer (A549 and MCF-7 cell lines), and antioxidant studies. Among all the compounds tested for antidiabetic potential, **237a**, **237b**, and **237c** exhibited substantial inhibition activity, with IC_50_ values of 20.12 ± 0.19 μM, 21.55 ± 0.46 μM, and 24.92 ± 0.98 μM, respectively, compared to the reference compound acarbose (IC_50_ = 12.68 μM) (see Figure 12).

Kaur et al. reported a series of novel isatin–pyrazole hybrids and tested their antidiabetic activity [113]. Among the tested compounds, **238** (Figure 12) appeared to be most potent with IC_50_ = 3.26 ± 0.25 μM, which was ~146-fold more potent than acarbose (*IC*_50_ = 478.07 ± 1.53 μM). Kausar and coworkers [114] synthesized Celebrex derivatives and investigated their antidiabetic effectiveness. Many of the evaluated compounds exhibited good activity. Compound **239** emerged as the most potent inhibitor of the *α*-glucosidase enzyme (IC_50_ = 92.32 ± 1.530 μM), comparable to the standard drug acarbose (IC_50_ = 875.75 ± 2.08 μM). Compound **240** exhibited excellent antidiabetic activity (*IC*_50_ = 5.8 µM) compared to the reference acarbose (*IC*_50_ = 58.8µM) and other evaluated compounds (**241**, *IC*_50_ = 65 µM and **242**, IC_50_ = 103 µM) under the same conditions [115]. Shen described a series of novel pyrazolo [1,5-*a*]pyrimidine derivatives as promising and selective DPP-4 inhibitors [116]. Compound **243** was 2-fold superior to alogliptin (IC_50_ = 4 nM) and notably selective over DPP-8 and DPP-9 (>2000-fold). The in vivo IPGTT assays in diabetics showed that **243** significantly lowers blood sugar by 48% at 10 mg/kg. The in vitro antidiabetic potential of compound **244** was evaluated against α-glucosidase and α-amylase enzymes. The results revealed that **244** with IC_50_ = 60.45 ± 1.23 μM showed superior α-glucosidase effectiveness relative to acarbose (IC_50_ = 89.12 ± 2.08 μM) [117]. Nevertheless, compound **244** was inactive against α-amylase (see Figure 12).

### 4.6. Antileishmanial

The incorporation of a heteroaromatic ring coupled with a 1,3,4-oxadiazole moiety improved the antileishmanial activity. Compounds **245, 246**, and **247** (Figure 13) proved the dose-dependent killing of the promastigotes with corresponding IC_50_ values of 33.3 ± 1.68, 40.1± 1.0, and 19.0 ± 1.47μg/mL, respectively [118]. Additionally, the compounds (**245, 246**, and **247**) displayed IC_50_ values of 44.2 ± 2.72, 66.8 ± 2.05, and 73.1 ± 1.69 μg/mL, respectively, on amastigote infectivity. These compounds depicted a comparable point in dose-dependent parasite killing with the standard drug, pentamidine (IC_50_ = 2.6 ± 0.32 μg/mL). Camargo et al. synthesized a series of novel pyrazole hybrids [119]. The hybrids were investigated in vitro against the promastigote of *Leishmania amazonensis*. At the same time, the hybrids were examined against the epimastigote of *Trypanosoma cruzi* (*T. cruzi*). The *S*-methyl thiosemicarbazones **248a**–**248c** and 2-amino-1,3,4-thiadiazole pyrazole hybrids **249a**–**249c** displayed significant antileishmanial and antitrypanosomal properties. The substitution of Br, OCH_3_, or NO_2_ at the *para* position of the aryl ring attached at position five of pyrazole favored their activity. The substituent attached to position three of the pyrazole ring also influenced the activity of the evaluated compounds. Silva et al. synthesized and screened a series of 1,5-biaryl 3-arylaminomethyl 4-carboxyethyl pyrazoles and screened against *L. amazonensis* and *T. cruzi* [120]. The most active compounds **249**, **250**, and **251** demonstrated similar profiles against both *L. amazonensis* and *T. cruzi* parasites, describing their dual activity. Meanwhile, compound **249** induced morphological and ultrastructural alterations in the promastigote of *L. amazonensis* (see Figure 13).

### 4.7. Antimalarial

Compound **252** (Figure 13) has been demonstrated to be an excellent inhibitor of Falcipain-2, with an IC_50_ value of 14 μM [118]. Akolkar et al. [121] have examined the antimalarial potency of **253**, **254**, and **255**. Compounds **253** and **254** (IC_50_ = 0.47 µM) were equipotent regarding efficacy compared to the standard drug quinine (0.83 µM). The inhibition activity of **255** (0.21 µM) was 4-fold superior to the standard drug. Molecular hybrids of the thiophene, pyrazoline, and benzene rings enhanced the antimalarial activity. Strašek and coworkers [122] reported the synthetic route of the tetrahydropyrazolo [1,2-*a*]pyrazole-1 carboxylates. The reactions yielded mixtures of 7-oxo-2,3,5,6 tetrahydropyrazolo [1,2–*a*]pyrazole-1-carboxamides **256** and **257**. An assessment inhibition of dihydroorotate dehydrogenase of *Plasmodium falciparum* (*Pf*DHODH) was demonstrated. The strongest potency was found in compound **257** with an IC_50_ value of 2.9 ± 0.3 μM). All the evaluated compounds developed selectivity for *Pf*DHODH more than *Hs*DHODH. Gogoi et al. [123] reported dimethoxy pyrazole 1,3,5-triazine derivatives as a novel class of potent antimalarial agents with good toxicity profiles. Within the series compound **258** was observed as a promising antimalarial agent (see Figure 14).

### 4.8. Antioxidant

Antioxidant activities of pyrazoline derivatives were screened using the 2,2,-diphenyl-1-picrylhydrazyl (DPPH) radical scavenging method. All the tested compounds showed antioxidant activity [88]. Compound **197c** bearing 4-fluoro and 4-methyl substituents was the most potent antioxidant agent among all the tested compounds at all the concentrations. Compound **234** has been reported as a promising antioxidant agent [109]. Compounds **237d** and **237c** have been revealed as having effective antioxidant activity (IC_50_ values = 4.25 μM and 5.40 μM, respectively) [112]. The results of an evaluation of synthesized thiazolidine-2,4-dione-pyrazole conjugates as antioxidant agents showed the efficacy of all the examined compounds [107]. Compounds **259a** and **259b** and **260** (see Figure 14) showed the most potent results, with IC_50_ values of 110.88, 127.18, and 128.55 μg/mL, respectively. The standard drug ascorbic acid showed an IC_50_ value of 81.12 μg/mL. The synthesis of functionalized pyrano [2,3-c]pyrazoles and pyrazolopyrano [2,3-d]pyrimidines containing a bioactive chromone moiety has been achieved, along with their antioxidant activity [124]. The in vitro antioxidant activity was determined using DPPH radical scavenging methods. Among the tested hybrids, **261–264** displayed promising activity with all the concentrations in the evaluation with the reference drug. The hybridization of pyranopyrazole with the pyrimidine moiety with substituted NH and OH groups improved the antioxidant properties. Ali et al. [125] reported the synthesis of a novel series of pyrazoline **269a**–**269e**, phenylpyrazoline **270a**–**270e**, isoxazoline **271a**–**271e**, and pyrazoline carbothioamide derivatives **272a**–**272e** using chalcones as a precursor **268a**–**268e**. The hybrid compounds were vetted for in vitro antioxidant activity using DPPH, nitric oxide (NO), and the superoxide radical scavenging (RSA) assay, along with 15-lipoxygenase (15-LOX) inhibition activity. Pyrazoline carbothioamide derivatives **272a** and **272e** were the most potent anti-LOX compounds, 2.2- and 2.1-fold superior to quercetin, while compounds **269a**, **270e**, **271b**, **271c**, **272a**, **272c**, and **272e** exhibited substantial RSA in all the three in vitro assays relative to the ascorbic acid, along with 15-LOX inhibition potency. The presence of electron-donating groups (CH_3_ and OCH_3_) or halogens (di-Cl) on the benzene ring enhanced the inhibition activity. The potential antioxidant activity of **272a** and **272e** were comparable in all three assays. Compounds **271b**, **271c**, and **271e** (Figure 15) showed significant in vivo antioxidant potential compared to the standard group at a dose of 100 mg/kg B.W. Meanwhile, there was an increase in CAT activity, the GSH level, and a decrease in lipid peroxidation in the treated rat liver compared to the control treatment. The in vitro antioxidant effectiveness of 4-(arylchalcogenyl)-1*H*-pyrazoles bearing sulfur or 1*H*-pyrazole groups has been investigated in different assays by Oliveira et al. [126], along with their oxidative stress impacts in biological systems. Compounds **273** and **274** showed significant inhibition in the ABTS assay, revealing that the mechanisms of the antioxidant action of compounds **273** and **274** were connected to their ability to donate electrons. Additionally, compounds **273** and **275** are more potent in the NO scavenging assay, while **274** reduced the lipid peroxidation levels in the brain and the liver after 72 h of treatment remarking on the compound efficacy in oxidative stress. A new series of pyrazole-containing heterocyclic skeletons—namely, pyrimidine, triazole, triazepine, pyrrolone, and thiadiazolopyrimidine—along with acylthiourea derivatives, were synthesized from 2-cyano-3-pyrazolylpropenoyl isothiocyanate by Badawy et al. [127]. The antioxidant screening of all the synthesized compounds showed that pyrimidinethione derivatives **276** and **277** were the most potent antioxidant agents. El-Borai et al. [128] achieved a biological evaluation of the cytotoxicity, antihemolytic and antioxidant activities of some thienopyrazole compounds. The antioxidant activity of the examined compounds was achieved by utilizing the DPPH radical scavenging assay with ascorbic acid as the reference. Compound **278** exhibited excellent radical scavenging activity, with an IC_50_ value of 4.49 μg/mL comparable to an IC_50_ of 4.76 μg/mL. The excellent antioxidant result was obtained due to the existence of the two amino groups on the pyrimidine ring. Additionally, **279** exhibited strong antihemolytic and antioxidants, justifying that the antioxidant activity may protect red blood cells from hemolysis. The insertion of chlorine atoms, hydroxyl, and cyanide with a pyrimidine ring in a single moiety enhanced the activity of **279**. In addition, **280** was noxious to all the tested cancerous cell lines; however, a lower cytotoxicity activity against the normal fibroblast cell line was observed. Elnagdy and colleagues [129] described a synthetic route to pyrazole analogs by using copper oxide nanoparticles (CuO-NPs) as catalyzed. The compounds were evaluated for their antioxidant activity using the DPPH radical scavenging assay. Most of the compounds tested demonstrated a greater interaction with the DPPH radical relative to the standard compound Trolox (IC_50_ = 11.48 mM). The compounds **281**, **282**, and **283** showed maximum antioxidant activity in the order of **282** > **281** > **283**, with IC_50_ values of 3.06, 3.53, and 5.42 mM, respectively. The ability of **281** and **282** to recover the DPPH radical assay resulted from the prolonged conjugation in compounds, while that of compound **283** was due to a phenolic hydroxyl group at the *ortho*-position and a fluorine group. The condensation reaction between 1,3-thiazole or aminopyridine derivatives and 1*H*-pyrazole,3,5-dimethyl-1*H*-pyrazole or 1,2,4-triazole was described by Kaddouri and colleagues [130]. The reaction produced novel heterocyclic compounds containing pyrazole, thiazole, and pyridine. Additionally, the DPPH scavenging assay was utilized to investigate their antioxidant activity. Ligand **284** showed the best antioxidant activity, with an *IC*_50_ value of 4.67 μg/mL, while the *IC*_50_ value for the reference compound was 2 μg/mL (ascorbic acid). The applicable route for the direct synthesis of (*E*)-ethyl 2-benzylidene-3-oxobutanoate through the 3 + 2 annulation method, including the investigated in vitro antioxidant vulnerabilities through the DPPH and hydroxyl radical scavenging methods of this compound, have been reported [131]. The assays showed that compound **285** has a strong antioxidant power (see Figure 16).

The multicomponent reaction of some heterocyclic compounds with activated acetylenic, alkyl bromides, triphenylphosphine, and hydrazine in water under ultrasonic irradiation yielded pyrazole derivatives in better yields [132]. Additionally, the antioxidant activities of the compounds were examined using DPPH radical scavenging and the ferric-reducing power assay. Compound **286** (Figure 16) exhibited exceptional DPPH radical scavenging activity and greater reducing power compared to the standard reference butylated hydroxytoluene (BHT) and 2-tertbutylhydroquinone (TBHQ). Compounds **287** and **288** have been reported as more potent antioxidant inhibitors than ascorbic acid and butylated hydroxyanisole (BHA) [133]. The corresponding IC_50_ values for **287** and **288** from the DPPH radical assay were 0.245 ± 0.01 and 0.284 ± 0.02 μM, respectively. These compounds have more potent RSA than ascorbic acid (IC_50_ = 0.483 ± 0.01 μM). In the hydroxyl radical scavenging assay, compounds **287** and **288** showed IC_50_ values of 0.905 ± 0.01 μM and 0.892 ± 0.01 μM, respectively. They displayed greater RSA than BHA (IC_50_ = 1.739 ± 0.01 μM).

Patil et al. [134] prepared sulfonic acid functionalized 1,4-diazabicyclo [2.2.2]octane assisted on Merrifield resin, [MerDABCO-SO_3_H]Cl as a catalyst to synthesized pyrazolopyranopyrimidines in one-pot four-component reactions in an excellent yield. The antioxidant effect of the synthesized compounds was determined using the 1,1-diphenyl-2- DPPH radicals scavenging assay, and ascorbic acid was used as a standard control. Among the evaluated compounds, **289–292** showed excellent antioxidant activity compared to the standard ascorbic acid due to the incorporation of an electron-withdrawing substituent (nitro group) on the phenyl ring, enhancing the resonance impact stabilizing the consistently formed radical. Compounds **293** and **294** have been reported as promising antioxidant agents [135] (see Figure 16).

### 4.9. Antituberculosis

Jagadale et al. [136] described the pathway to synthesize thiazolyl-pyrazolyl-1,2,3-triazole derivative **295** and bis-pyrazolyl-1,2,3-triazole **296** derivative, along with their antimycobacterial activity against *M. tuberculosis* (*Mtb*) with H37Ra dormant and active. The antimycobacterial activity revealed that most of the compounds showed moderate to good activity against both strains of M. tuberculosis. Compounds **295a**–**295c** and **296a** and **296b** exhibited good activity against the *Mtb* H37Ra active strain; also, compounds **295d** and **295e** and **296c**–**296e** displayed good activity against the *Mtb* H37Ra dormant strain. Using Pd/Cu catalyst coupling-cyclization strategy, 3-indolylmethyl substituted (pyrazolo/benzo) triazinone derivatives have been expediently prepared in a one-pot reaction [137]. The synthesized compounds were tested for chorismate mutase (CM) inhibitory properties in vitro using an assay that measured the enzyme’s catalytic activity (MC) in converting chorismate (substrate) to prehenate. The best active compounds, **297** and **298**, showed 78% inhibition at 30 μM. Meanwhile the concentration-dependent evaluation resulted in IC_50_ values of 0.40 ± 0.05 µM and 0.85 ± 0.10 μM for compounds **297** and **298**, respectively. Compound **299** has also been shown to maintain good potency against clinical samples from the four main lines and strains containing isoniazid or rifampin resistance mutations [138]. The mutated strains in MmpL3 were resistant to **299** and under replication conditions, and it exhibited bactericidal activity against *Mtb*. However, compound **299** was not effective in an acute model of tubercular infection. This is likely the result of in vivo exposure remaining above the minimum inhibitory level for a restricted period (see Figure 17).

Hu and coworkers [139] reported novel series of pyrazolo [1,5-a]pyridine-3-carboxamide (PPA) conjugates bearing diaryl side chains and their antitubercular activity. Most of the evaluated compounds were highly potent in vitro against *Mtb* strains, including H37Rv (MIC = < 0.002–0.381 μg/mL), rINH (MIC = < 0.002–0.465 μg/mL), and rRMP (MIC = < 0.002–0.004 μg/mL). Notably, compound **300** demonstrated favorable in vitro activity against *Mtb* H37Rv, rRMP, and rINH, with corresponding MIC values ≤ 0.002 μg/mL and a lack of toxicity against Vero cells. Moreover, in vivo studies showed that **300** substantially lessened the mycobacterial load in a mouse model infected with H37Ra. Other reported antituberculosis pyrazole derivatives and the corresponding references are shown in Table 1.

### 4.10. Agrochemical

Series of novel pyrazole−isoindoline-1,3-dione hybrids as favorable 4-hydroxyphenylpyruvate dioxygenase (HPPD) inhibitors were designed by combining 2-benzoylethen-1-ol and isoindoline-1,3-dione into a single moiety [153]. Among the evaluated compounds against *Arabidopsis thaliana* HPPD in vitro, using mesotrione and pyrasulfotole as the positive control, the IC_50_ of **316** (Figure 18) was extended to 90 nM. In addition, **316** was identified as the most promising inhibitor, with a *K*_i_ value of 3.92 nM, which was ~10 times superior to pyrasulfotole (*K*_i_ = 44 nM) and 300 times marginally superior to mesotrione (*K*_i_ = 4.56 nM). Jiang et al. [154] designed and synthesized novel heptacyclic pyrazolamide conjugates using the scaffold hopping approach. The insecticidal activities of all synthesized compounds were examined against *P. xylostella* in vivo at 500 mg/L. Meanwhile, the marketed insecticide—namely, tebufenozide—was used as the reference drug. Compounds **317** and **318** flaunted excellent insecticidal activities (>90%) against *P. xylostella*. Additionally, compound **317** displayed 100% insecticidal activity at the dose of 200 mg/L. The lower dose and LC_50_ value of **317** (64.13 mg/L) was akin to tebufenozide (LC_50_ = 33.83 mg/L). Zhao et al. [155] reported a novel series of fluoro-substituted compounds bearing altered pyrazole and their anti-larvicidal effects. The larvicidal activity unveiled fluoro-substituted compounds to have good to excellent activities against *M. separata* and *P. xylostella*. The corresponding LC_50_ values for **320a** and **320b** against *P. xylostella* were 2.9 × 10^−6^ mg/L and 3.1 × 10^−6^ mg/L, respectively, superior to the LC_50_ of chlorantraniliprole (4.6 × 10^−5^ mg/L). In addition, fluoro-substituted compounds **320a**–**320c** bearing ether groups at position three of the pyrazole showed better inhibitory effects than compounds with halogen, amide, or ester groups substituents. The insertion of fluorine atoms on the ethoxy group enhanced the larvicidal activity. Compound **320a** exhibited the 50% larvicidal mortality against *M. separata* to 0.1 mg/L. Moreover, **320a** displayed 90% larvicidal activity against *P. xylostella* at 10^−5^ mg/L, higher than that of chlorantraniliprole. Judge and colleagues [156] revealed substituted 3-hydroxyprozole derivative **321** as a promising herbicidal agent. Pyridylpyrazole-4-carboxamides bearing 1,3,4-oxadiazole rings were designed and synthesized by dehydration of aromatic hydrazine derivatives and formanilides [157]. The synthesized compounds were further evaluated for their insecticidal activities (*Plutella xylostella*). Among the examined compounds, **322** displayed promising activity as follows, 67%, 50%, 34%, 20%, and 17% activity at the concentrations of 100, 50, 10, 5, and 1 μg/mL, respectively (see Figure 18).

## 5. Conclusions

Pyrazoles are five-membered heterocyclic compounds containing nitrogen. They are an important class of compounds for drug development; they constitute an essential class of hit compounds to develop new pharmacological agents to treat various infections of clinical primacy. With such a diverse range of biological activities, they have attracted much attention from researchers focusing on synthesizing different pyrazole analogs to developing novel and more effective drugs. This literature review documented various synthetic pathways to pyrazole derivatives and the biological potential of some pyrazole derivatives in recent years. Their biological activity properties, such as antibacterial, analgesic, anti-inflammatory, anticancer, antibacterial, antidiabetic, antioxidant, and agrochemical, were detailed in this review. The information presented in this review will assist prospective researchers in further investigating pyrazole derivatives and update scientists with promising biological activities of recently developed derivatives. Additionally, this will allow them to identify other derivatizations that can be explored. However, where the pyrazole unit itself plays a significant role in the compound’s mode of action, including cases where the pyrazole is more of a structural element, still needs to be explored. Additionally, the molecular hybridization of pyrazole with other bioactive compounds will be explored in our future work.

## Data Availability

Not applicable.

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
