# Peer review of "A Review of the Recent Development in the Synthesis and Biological Evaluations of Pyrazole Derivatives"

_biomedicines, 2022, doi:10.3390/biomedicines10051124_

Round 1

Reviewer 1 Report

The manuscript fails to add any added value or insight and is a simple enumeration of the scientific literature. In its current form I recommend rejection and re-submission at a later stage, as the changes needed are likely too large for a revision.

The first part can best be described as a listing of every single conceivable way to prepare pyrazoles, listing dozens of synthetic strategies without ever venturing in a rationalization. A simple listing of literature has no real benefit for the reader. The aim of a review should be to provide insight, e.g. to compare methodology or to rationalize synthetic strategies. In its current form there is just a non-structured overview. For this review to benefit the scientific community, the authors should reconsider their approach and provide a better structure to the manuscript. For example, group reactions based on reaction type, or per reagent class, or per substitution pattern of the formed pyrazole. Additionally, rather than just providing 2-5 lines of a paraphrased abstract for each different paper, reactions should be compared, major findings should be mentioned, also relative to previous or alternative methods. What is the scope of the methods? Are there discrepancies between published materials? For selected non-trivial cases reaction mechanisms could be detailed. Etc.

The second part of the manuscript ventures into a long listing of biological properties, again without providing rationale. I believe the goal of the authors, namely, to provide an overview of all biological properties of pyrazole-containing molecules is simply too ambitious. The scaffold is ubiquitous and therefore basically would result in having to conclude the scaffold is relevant for every disease area. Here it might be better to just provide a few well-structured and rationalized examples. For example, cases where the pyrazole unit itself plays a major role in the mode of action of the compound and cases where the pyrazole is more of a structural element, with side-by-side comparisons of analogous non-pyrazole compounds and the effect of the incorporation of a pyrazole. The authors could for example also provide an overview of why this scaffold is so important in drug design (how does it influence the polarity, hydrogen bonding network, toxicity, solubility, etc.). This type on non-exhaustive description of the biological and medicinal chemistry properties would likely be much more beneficial for the reader than a generic listing of all recent applications of the pyrazole building block.

Finally, the authors should review the English. While several sections are well-written others are littered with grammatical errors.  

Reviewer 2 Report

This is "A review of the recent development in the synthesis and biological evaluations of pyrazole derivatives" focuses on a concise overview of pyrazole pharmacophores’ synthesis and biological activities. The work is quite nicely organised and indeed could serve as a reference guide for researchers interested in the field as long as some corrections, additions and language editing are appropriately made.

  • Please ensure that all subscripts are correctly shown in the whole text such as IC50 LC50 O2 etc. Keep the same format.
  • Is it an option to refer to Literature from 2016-2021?? it is necessary to describe the research methodology followed for this review, which search engines where used, for what years etc.
  • Please check spaces before [ref], between nambers and units etc
  • Figure A + B - they are referred to in the introduction but they do not exist
  • Figure 1 Legend - Not correctly placed. Probably refers to Scheme 1
  • Scheme 1 - compds 17 & 18 yield 0%?? please explain and discuss in the text
  • Scheme 4 - is it 29i-l or 28 i-l??? this scheme should be redrawn to save space by drawing pyrazoline-R, where R = Ph, p-FPh, p-MePh etc. The same should be done to all schemes where applicable, such as for 33d-33g keep the general R4 and explain 33d R = p-ClPh etc 
  • Please add the synthetic methodology described in the legend of all scheme i.e ......via CuCl etc
  • structure 33d there is no R1
  • MMT K10 for the synthesis - please make a small comment on the nature and the characteristics of the material used. Also, add some comparison comments such as the use of Cu(OH)and CuCl etc. Overall, the synthetic procedures are presented as bullets with no connection or comparison between them. This should be improved.
  • For the biological activities - % inhibition of what?? Selectivity Index between what???  Line 200- what is IR??? Line 208 - IC50 in what??? it is necessary to state the experimental set up and target when results are presented. One of way of dealing with that is if the COX-2 inhibitors are pulled together and discussed together. Then the selectivity over COX-1, then the paw edema active compounds. It would be advisable to start with a short description of the anti-inflammatory potential of studying these targets and then the results with some kind of SAR where applicable.
  • Line 324 - significant toxicity G150 - 15 mM is it correct or is it μM??
  • Line 327-328 - there only numbers with no commas, no connection with the next sentence. Please correct/rephrase accordingly
  • lINE 448 - against which cell lines???
  • Section 2.3 - Antibacterial-Antifungal
  • Line 700 - express in µM

In the anticancer section some more mechanistic targets of the compounds should be added. 

In general, the text should become more solidly interconnected not just a bullet point presentation of results and molecules.

Reviewer 3 Report

The review is devoted to the synthesis and biological evaluation of pyrazole derivatives. The topic of the review is very important and interesting, since the pyrazole ring is a component of many drugs. The review contains the very latest information on this kind of heterocycles. Unfortunately, the paper contains many drawbacks.

  1. Because the data on pyrazoles are vast and difficult, if not impossible, to describe in a single review, the authors needed to indicate how many years the literature was collected.
  2. The review lacks fundamental reviews on this class of heterocycles, for example, Comprehensive HeterocyclicChemistry I-IV and many others.
  3. The review, in part of methods for the synthesis of pyrazoles, is completely unstructured;it is necessary to analyze this material and divide it into appropriate parts.
  4. Part of the material on page 2 is simply omitted.
  5. Titles for a number of schemes (1, 4, 7-9, 23, 25, 26, etc.) are too general and often coincide with each other and do not reflect the specific essence of this scheme.
  6. Most huge schemes can be easily reduced by introducing the meanings of the radicals, excluding the formulas of the final pyrazoles.Schemes 25 and 26 can be combined.
  7. There are many errors and misprints in the paper; the paper requires careful editorial revision, including the list of references.
  8. The biological activity part of the review is often a listing of data published in the literature without analyzing them;phrases are often repeated in the text.

The paper can be recommended for publication in the journal ‘Biomedicines’ after major revision.

Reviewer 4 Report

Paper is a systematic review regarding synthesis and biological properties of pyrazole derivatives.

Surely the authors have done a lot of bibliographic work, but in this version, paper can’t be accepted, and major revisions are necessary.

In particular:

1) in the abstract authors have to specify the time lapse of bibliographic research and the databases used.

2) introduction is too long: please delete lines 28-29; line 36: please change “Figure B” with “Figure 1”.

3) Figure 1: please delete compounds in which pyrazole nucleus is fused with other cycles as Formycin Fluviol and others. In the same figure compounds as celecoxib, mavacoxib, deracoxib reported in scheme 27 should be added

4) in all text synthesis and biological evaluation of fused-pyrazoles (compounds 41, 45, 52, 54, 59, 66, 152, 153, 157, 158, 159, 189, 190, 204, 205 and many others) or pyrazoline derivatives (22, 25, 187, 197, 203 and many others) should be deleted. On the contrary, other pyrazoles recently reported should be added (for example M.G. Signorello, New Series of Pyrazoles and Imidazo-pyrazoles Targeting Different Cancer and Inflammation Pathways” Molecules, 2021, 26, 5735. Doi: 10.3390/molecules26195735; S. Alfei, et al. Pyrazole-Based Water-Soluble Dendrimer Nanoparticles as a Potential New Agent Against Staphylococci” Biomedicines, 2021, 10(1):17. doi: 10.3390/biomedicines10010017; E. Morretta, Synthesis, functional proteomics and biological evaluation of new 5-pyrazolyl ureas as potential anti-angiogenic compounds” Eur J Med Chem. 2021, 226, 113872. doi: 10.1016/j.ejmech.2021.113872)

5) please check the structure of compounds 154 and 164.

6) in the caption of reaction scheme, as well as in the caption of some figures, please specify the type of reported pyrazoles (for example pyrazole 3-substituted or similar).

7) in the caption of reaction scheme please insert corresponding references.

8) please insert in all references the abbreviation of journal type.

9) please change IC50 with IC50 (subscript).

10) please check english language.

Round 2

Reviewer 1 Report

While the authors have addressed some of the shortcomings, most importantly a structured overview of the synthetic approaches, the overall quality has not sufficiently been improved to warrant publication in the current form.

For example in figure 2, there are several errors where not all double bonds are shown in the structures of Lonazolac and Sildenafil. Furthermore Sildenafil is the active pharmaceutical ingredient in Viagra. The structure of "viagra" next to Sildenafil is wrong.
There are still several ambiguous statements and awkward sentences for example:
"when alkyl-based sulfonyl hydrazine was incorporated, the reaction was not positive" (If it is incorporated the reaction must have worked, since now it's part of the scaffold. Then what is a reaction that is not positive? Poorly yielding, or not proceeding, ...?)
"the salt formation was then performed with a porridge of the free base and citric acid"
etc.
Please revise this section thoroughly.

The biological activity section of the manuscript remains difficult to read due to the lack of a structured approach. The authors indicated that they wish to maintain their approach to refer to all studies in the 2018-2021 time frame rather than providing a few key examples. This is of course their prerogative, but an additional degree of structuring and rationalization would make this much easier to read. That being said, the level of writing of this biological activity section is much better then that of the chemistry section.

Finally, the conclusion needs a lot of work. As indicated before, the aim of a review should be to provide insight and guidance. So far, guidance is missing from the manuscript. The conclusion section is ideally suited for the authors to provide an opinion on the future evolution of the field; to indicate future directions that they feel warrant extra attention; to highlight gaps in the current knowledge; etc. Please revise.

Author Response

Reviewer 1#

We appreciate the reviewer's time to review our manuscript and for all the critical comments, observations, and suggestions raised to improve our manuscript. All the comments, observations, and suggestions have been answered below.

For example in figure 2, there are several errors where not all double bonds are shown in the structures of Lonazolac and Sildenafil. Furthermore Sildenafil is the active pharmaceutical ingredient in Viagra. The structure of "viagra" next to Sildenafil is wrong.

Thank you for the observation. This was a mistake from the authors; therefore, we have rechecked all the structures and made necessary corrections.

There are still several ambiguous statements and awkward sentences for example:
"when alkyl-based sulfonyl hydrazine was incorporated, the reaction was not positive" (If it is incorporated the reaction must have worked, since now it's part of the scaffold. Then what is a reaction that is not positive? Poorly yielding, or not proceeding, ...?)

Thank you for the observation; alkyl sulfonyl is different from aryl sulfonyl; however, we have rephrased the sentence.

Lines 51-58

The tandem reactions between amine-functionalized enaminones 1 and aryl sulfonyl hydrazine or tosylhydrazone derivatives 2, 3 in the absence of metal catalyst have been reported. The synthesis of the substituted pyrazoles 4, 5 occurred in water, TBHP, and NaHCO, respectively. In addition, when alkyl-based sulfonyl hydrazine such as methyl sulfonyl hydrazine was incorporated, the reaction was not successful. Besides, when the reaction was carried out in EtOH and DMF.

"the salt formation was then performed with a porridge of the free base and citric acid"
etc.
Please revise this section thoroughly.

Corrected

The biological activity section of the manuscript remains difficult to read due to the lack of a structured approach. The authors indicated that they wish to maintain their approach to refer to all studies in the 2018-2021 time frame rather than providing a few key examples. This is of course their prerogative, but an additional degree of structuring and rationalization would make this much easier to read. That being said, the level of writing of this biological activity section is much better then that of the chemistry section.

Finally, the conclusion needs a lot of work. As indicated before, the aim of a review should be to provide insight and guidance. So far, guidance is missing from the manuscript. The conclusion section is ideally suited for the authors to provide an opinion on the future evolution of the field; to indicate future directions that they feel warrant extra attention; to highlight gaps in the current knowledge; etc. Please revise.

We appreciate the reviewer for the observation and comments.

We have revised the conclusion as you rightly suggested

Pyrazoles are five-membered heterocyclic compounds containing nitrogen. They are an important class of compounds for drug development; they constitute an essential class of hit compounds to develop new pharmacological agents to treat various infections of clinical primacy. With such a diverse range of biological activities, they have attracted much attention from researchers focusing on synthesizing different pyrazole analogs to develop novel and more effective drugs. This literature review has documented various synthetic pathways to pyrazole derivatives and the biological potential of some pyrazole derivatives in recent years. Their biological activity properties such as antibacterial, analgesic, anti-inflammatory, anticancer, antibacterial, antidiabetic, antioxidant, and agrochemical were detailed in this review. The information presented in this review will assist prospective researchers in further investigating pyrazole derivatives and update scientists with promising biological activities of recently developed derivatives. Besides, this will allow them to identify another derivatization that can be explored. However, where the pyrazole unit itself plays a significant role in the compound's mode of action. Including cases where the pyrazole is more of a structural element still needs to be explored. Also, the molecular hybridization of pyrazole with other bioactive compounds will be explored in our future work.

Reviewer 2 Report

Accept the manuscript in the present form as the authors have addressed all the points and concerns raised and improved the manuscript immensely.

Author Response

We thank the reviewer for the positive assessment of our work.

Reviewer 3 Report

The authors took into account almost all the comments of the reviewers and the review can be recommended for publication after editorial corrections.

Author Response

(The authors gave the same response as above.)

Reviewer 4 Report

The major part of my suggestions has been followed, but some minor corrections are necessary:

1) compounds 125 and 180 are pyrazoline derivatives and should be delete;

2) compound 243 is not pyrazole derivative.

3) in compounds 186 R4 is not present, please check.

4) in compounds 188 R and R1 are inverted, please check.

Author Response

We thank the reviewer for the positive assessment of our work. The remaining minor corrections suggested by the reviewer have been implemented in the revised manuscript.